# cyCombine allows for robust integration of single-cell cytometry datasets within and across technologies

Christina Bligaard Pedersen [1,2], Søren Helweg Dam[1], Mike Bogetofte Barnkob [3], Michael D. Leipold [4], Noelia Purroy[5,6], Laura Z. Rassenti[7], Thomas J. Kipps[7], Jennifer Nguyen[8], James Arthur Lederer [8], Satyen Harish Gohil[5,9,10], Catherine J. Wu [5,11] & Lars Rønn Olsen [1✉]

Combining single-cell cytometry datasets increases the analytical flexibility and the statistical power of data analyses. However, in many cases the full potential of co-analyses is not reached due to technical variance between data from different experimental batches. Here, we present cyCombine, a method to robustly integrate cytometry data from different batches, experiments, or even different experimental techniques, such as CITE-seq, flow cytometry, and mass cytometry. We demonstrate that cyCombine maintains the biological variance and the structure of the data, while minimizing the technical variance between datasets. cyCombine does not require technical replicates across datasets, and computation time scales linearly with the number of cells, allowing for integration of massive datasets. Robust, accurate, and scalable integration of cytometry data enables integration of multiple datasets for primary data analyses and the validation of results using public datasets.

[1] Department of Health Technology, Technical University of Denmark, Kongens Lyngby, Denmark. [2] Center for Genomic Medicine, Rigshospitalet—Copenhagen University Hospital, Copenhagen, Denmark. [3] Centre for Cellular Immunotherapy of Haematological Cancer Odense (CITCO), Department of Clinical Immunology, Odense University Hospital, University of Southern Denmark, Odense, Denmark. [4] Human Immune Monitoring Center, Institute for Immunity, Transplantation, and Infection, Stanford University School of Medicine, Stanford, CA, USA. [5] Department of Medical Oncology, Dana-Farber Cancer Institute, Boston, MA, USA. [6] AstraZeneca, Waltham, MA, USA. [7] Division of Hematology-Oncology, Department of Medicine, Moores Cancer Center, University of California, San Diego, La Jolla, CA, USA. [8] Department of Surgery, Brigham and Women's Hospital, Harvard Medical School, Boston, MA, USA. [9] Department of Academic Haematology, University College London, London, UK. [10] Department of Haematology, University College London Hospitals NHS Trust, London, UK. [11] Broad Institute of MIT and Harvard, Cambridge, MA, USA. ✉email: lronn@dtu.dk

Protein expression-based single-cell cytometry has evolved immensely over the past decades. While flow cytometry remains a staple of both basic cell biology research and clinical diagnostics[1], the introduction of mass cytometry (CyTOF) in 2009 increased the potential number of simultaneously measured markers to more than 45[2] as issues with signal spillover between reporter molecules and autofluorescence of cells were minimized[3,4]. More recently, spectral flow cytometry enables the measurement of 40 features or more without compromising throughput[5]. Sequence barcoding-based cytometry, such as CITE-seq, has even further increased the number of markers to the hundreds by almost completely eliminating signal spillover[6], and single-cell mass spectrometry is promising to increase feature counts even further[7–9]. Common to all these technologies is the desire to integrate data from different experiments, whether seeking to validate results using external datasets or aiming to increase the breadth and/or depth of the dataset used for a given study. This is rarely directly possible due to technical variance arising from data being generated with different antibody panels, reagent lots, or instruments; at different times; by different operators; etc.[10]. The resulting technical variance is commonly referred to as batch effects, and removing this undesired variance has remained a major unsolved challenge.

While many proposed methods offer means to alleviate the problem, the majority are designed for very specific applications, requiring technical replicates to be included across all batches, only enabling correction of batch effects in samples belonging to specific conditions, or being designed to work only on a specific type of cytometry data. These limitations preclude large-scale integration of data from different experiments, a feature that has become increasingly desired as more and more data is being published.

In this work, we have developed the cyCombine method for integration of cytometry data to overcome these challenges. We show that cyCombine enables quantifiably accurate harmonization of cytometry datasets, by removing the technical noise between batches, while maintaining the biological signal. We developed cyCombine to be independent of technical replicates across batches, as well as robust enough to harmonize cytometry data generated with different technologies.

## Results

**The cyCombine batch correction module**. The main engine of the cyCombine batch correction module is the tried and true empirical Bayes method for removal of batch effects, ComBat[11]. ComBat was first introduced in 2007 as a tool to address batch effects in DNA microarray data, but the empirical Bayes model has since proven useful for different types of bulk expression data. However, ComBat is not directly applicable to single-cell data, as it is designed to detect and remove technical variance between samples from different batches, while preserving biological variance between samples belonging to homogeneous conditions. However, in single-cell cytometry data, each sample is often characterized by vast heterogeneity in the expression patterns of the different cell types, thus prohibiting explicit modeling of technical and biological variance between samples.

In the cyCombine batch correction module, we address the intra-sample heterogeneity by considering each cell as its own sample and minimize the batch effects for groups of similar cells, one group at a time. The grouping of similar cells is done using a self-organizing map (SOM)[12], with an $8 \times 8$ node grid. This means that the cells will initially be clustered into 64 categories. This will typically be enough to capture the diversity of peripheral blood mononuclear cells, while ensuring that enough cells to capture the biological variance among cells from the same

batches, as well as the technical variance between batches are assigned to each cluster. The grid size can be adjusted if less or greater heterogeneity is anticipated. Generally speaking, we would advise to err on the side of overclustering, as long as the data set is of sufficient size. This will not negatively affect the performance of cyCombine, but will increase runtimes (for full discussion and examples see https://biosurf.org/cyCombine). In order to ensure that phenotypically similar cells cluster together across different batches, the expression of each marker is initially standardized within each batch. This is done either by transforming the expression values to Z-scores, which works well for fairly low-variance batches (e.g., data from different batches in an experiment), or ranks, which works well for high-variance batches (e.g., data stemming from different experiments or technologies). The transformed data are then used to cluster the cells using the SOM, and the node labels are assigned to the original expression value cells (Fig. 1a).

**The cyCombine panel merging module**. To integrate data from experiments designed with multiple panels of antibodies for increased feature breadth, cyCombine includes a module for panel integration. This module is likewise based on SOM clustering of cells from the different panels using the overlapping markers, followed by probability-based imputation of missing channels by drawing expression values from multi-dimensional kernel density estimates calculated on the cells from the opposing panel (Fig. 1b). The clustering and multidimensional draws ensure that co-expression patterns and frequencies of subtypes are maintained and only "true" cell types are imputed (see Supplementary Discussion).

**cyCombine enables large-scale integration of multi-batch, multi-panel cytometry data**. In order to demonstrate that cyCombine enables co-analysis of data from different experimental batches, we generated a CyTOF dataset consisting of 128 samples, run in seven batches. The experiment contained two conditions: 20 healthy donor (HD) samples and 108 chronic lymphocytic leukemia (CLL) samples, collected from 56 patients at two different time points. Samples were depleted of B cells in order to isolate and study the phenotypes of the non-malignant immune cells. Each sample was split in two and stained with two different antibody panels, overlapping by 15 markers and differing by 40 markers (Supplementary Data 1).

First, batch effects were minimized in each panel, after which batch effects of the 15 overlapping markers between the two panels were minimized (Fig. 2a, b and Supplementary Figs. 1 and 2). Then, the two panels were merged by imputing expression data from the non-overlapping markers. The integrated dataset consisted of 12,858,678 cells and the expression of 55 markers. The combined dataset was clustered based on a subset of 23 lineage markers using a SOM[12] and ConsensusCusterPlus[13] to 45 meta-clusters, which were labeled manually, merged, and cleaned-up into a total of 29 clusters (Fig. 2c and Supplementary Fig. 4). The percentage of cells from each sample assigned to each cluster correlated very strongly (Pearson correlation coefficient = 0.9996) between cells derived from the two panels. For both of the two panels, the batch correction resulted in an earth mover's distance (EMD) reduction of 0.66. Biological variance was retained in both panels, as indicated by the median absolute deviation (MAD) score between pre-batch and post-batch correction samples being 0.02 for both panels, and as shown in Supplementary Fig. 3, rare clusters are maintained after correction.

Within the 29 clusters we identified a range of T, NKT, myeloid, and NK cells populations (Fig. 2c and Supplementary Fig. 4). Interestingly, we observed that the proportion of the T

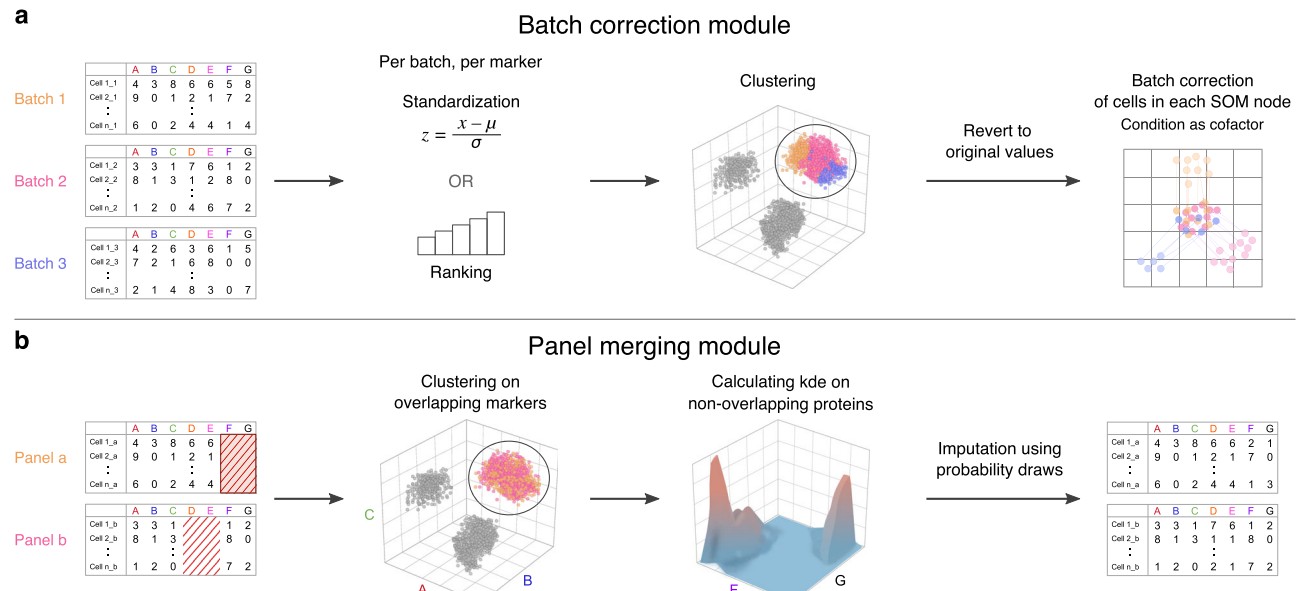

**Fig. 1 cyCombine overview. a** Batch correction workflow. First, expression values are transformed in each batch to enable co-clustering of samples from all batches. After clustering, the transformed values are reverted to expression values and ComBat is applied to each self-organizing map (SOM) cluster. **b** Panel merging workflow. Clustering is performed on overlapping markers, and the missing values for each cell in a panel are imputed using probability draws from the kernel density estimates (kde) from co-clustered cells of the other panel.

and NKT cell compartment was increased in CLL patients (Fig. 2d), as were circulating stem cells (as identified by CD34+ expression), especially closer to treatment (Fig. 2e), suggesting marrow stress with higher disease burden. In keeping with previously published data[14–16], we saw a decrease in naive CD8+ T cells, with corresponding increase in the CD8+ terminally differentiated effector memory (TEMRA) population when comparing close-to-treatment CLL samples to HDs (Supplementary Fig. 5). The use of HLA-DR in the staining further identified groups of CD8+ and CD4+ effector memory T cells that increased between CLL time point 1 and 2 with the CD4+ cluster being specifically enriched for PD-1 (Fig. 2f, g and Supplementary Fig. 5), similar to that reported by Elston et al.[15]. See also Supplementary Discussion.

**cyCombine removes technical variance and maintains biological variance.** Another scenario where batch correction is necessary is for the integration of external datasets. This is relevant when validating findings in public datasets or when performing meta analysis of multiple existing datasets. To demonstrate cyCombine's capability to handle integration of data generated in different experimental setups, we integrated CyTOF samples from two different datasets. The two datasets were generated at different facilities, on different versions of the CyTOF instrument, with different panels of antibodies conjugated to different isotopes. Applying cyCombine reduced the EMD by 0.76, making the two datasets directly comparable, and with an MAD score of 0.04, indicating minimal loss of biological variance. As a testament to the robustness of cyCombine, one dataset being B cell depleted did not affect the batch correction, nor did the correction introduce B cells into the depleted batch (Fig. 3).

When studying Fig. 3, it is noticeable that a small cluster (0.5%) appears in the Dana-Farber Cancer Institute (DFCI) set in the same UMAP position as the B cells from the Human Immune Monitoring (HIMC) set (11.9%). We do not expect B cells in the DFCI set, so one could suspect that this means that B cells have been artificially introduced by cyCombine. However, when looking closer at these

cells it becomes evident that their marker expression before correction is actually distinctly CLL cell-like, although with low CD19 expression explaining their presence after depletion. This fits with 82% of these cells originating from the CLL sample. While this observation makes biological sense, it highlights an important challenge when integrating cytometry: the breadth of the integrated dataset is limited by the overlapping markers in the two panels. In this example, the CLL cells are mislabeled as myeloid due to lack of the CD5 marker for CLL cells and corresponding lack of typical myeloid markers such as CD11b.

**cyCombine enables cross-platform data integration.** As cyCombine is agnostic to marker distributions, it enables integration of datasets generated on entirely different platforms. This can be highly useful in cases where different single-cell technologies have been applied to assess the same samples and one wishes to directly integrate the results. It is also possible to integrate data from different studies, even when the data was generated using different technologies. To demonstrate this feature, we applied cyCombine to three healthy donor peripheral blood mononuclear cell (PBMC) samples generated by CyTOF (HIMC dataset), CITE-seq (Illumina dataset), and spectral flow cytometry (Park et al. dataset[5]), respectively. While the raw data from the three data types assume distinct groupings in UMAP space (Fig. 4a), batch correction using cyCombine makes the data directly comparable (Fig. 4b). The resulting EMD reduction was 0.69 (Fig. 4c) and the MAD score 0.07. The clustering, subpopulation labeling, and marker expression of cells indicates that data are comparable only after correction (Fig. 4d–e and Supplementary Fig. 6).

**cyCombine scales linearly with the number of cells.** Another desirable application of cyCombine is for integration of very large cytometry datasets, e.g., from clinical trials or retrospective data from clinical diagnostics. Both the computation time and the memory requirements of cyCombine scale linearly with the number of cells and features, and, for example, the correction of 15 markers measured on 12,858,678 cells across two panels ran in 7 min on a standard laptop and required 10 GB of memory. This

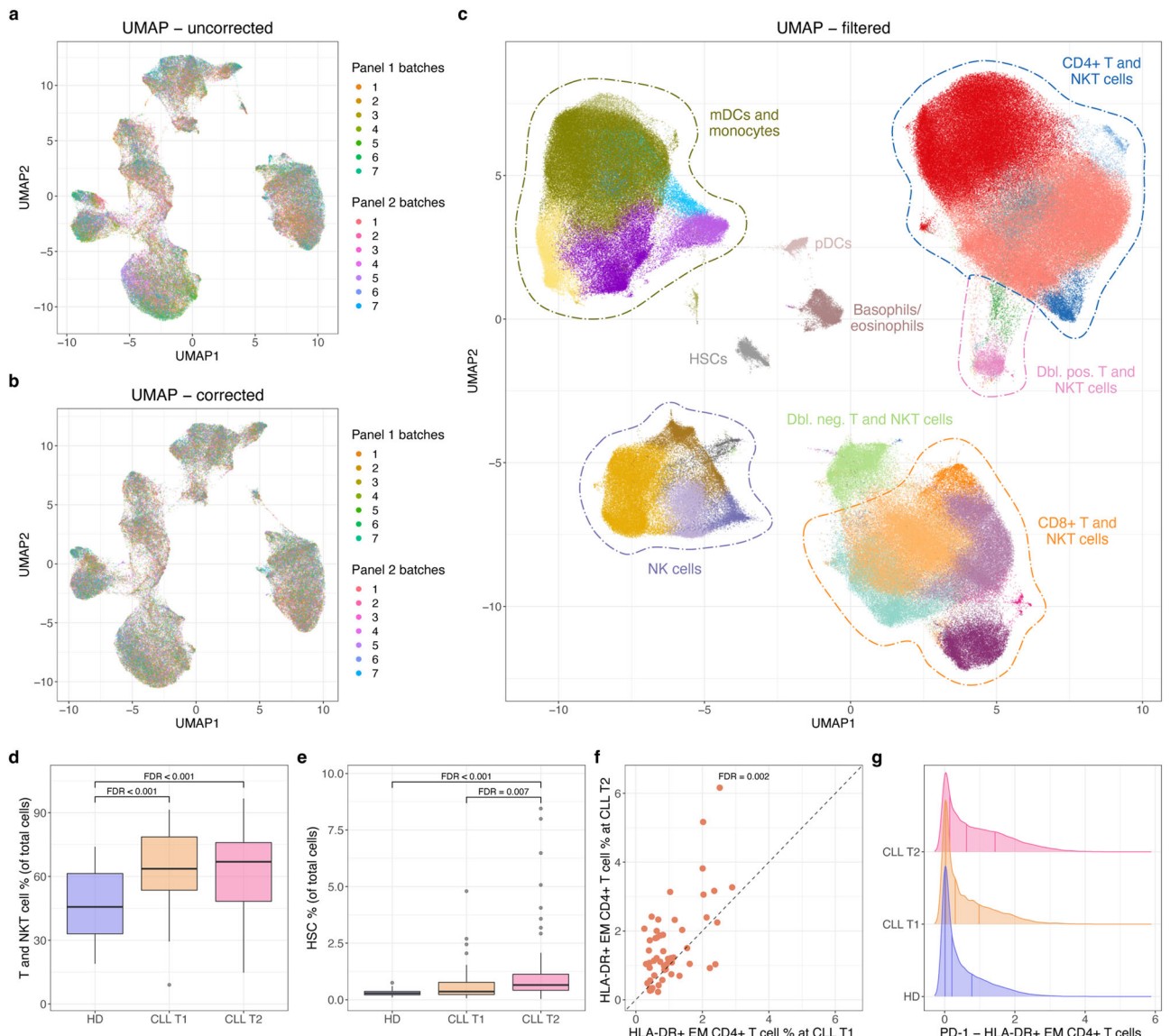

**Fig. 2 Integration and analysis of 128 CyTOF samples from seven different batches and two different panels. a** UMAP-based on expression of the 12 overlapping lineage markers included in the final clustering for both panels 1 and 2 before any batch correction. Using ~100,000 cells with equal sampling from all batches. **b** Same as in **a**, but after batch correction both within and between batches. **c** UMAP for up to 4000 cells from each of the 128 samples based on expression of the 23 clustering markers after removal of B, chronic lymphocytic leukemia (CLL), and poor-quality cells. Generated after panel merging, clustering, and filtering, detailed labels in Supplementary Fig. 4. **d, e** Box plots comparing the cell type proportion of two overall cell types between three sample groups: Healthy donor (HD) ($n = 20$), CLL time point 1 (T1) ($n = 52$), and CLL time point 2 (T2) ($n = 56$). The box plots show the medians (solid line in boxes), 25th and 75th percentiles as lower and upper hinges of the boxes, and whiskers extend to the furthest data point within 1.5* interquartile range from the hinges. Data points beyond this threshold are shown as circles. False discovery rates (FDRs) for the differential abundance testing are added to the comparisons yielding significant (FDR < 0.01) results. Please note the use of different y axes. **f** Scatter plot for the proportion of HLA-DR + effector memory (EM) CD4+ T cells in paired CLL T1 and T2 samples. FDR value from differential abundance testing within the T and NKT cell compartment. **g** Density plots for PD-1 expression levels in the HLA-DR + EM CD4+ T cell population (panel 1 cells only) for the three sample groups: HD, CLL T1, and CLL T2.

means that, while the memory requirements necessitate the use of a high performance computer, cyCombine can be applied to billions of cells in less than a day, and there is theoretically no limitation on the number of different datasets that can be integrated (for full runtime analysis see Supplementary Fig. 7 and Supplementary Discussion).

**cyCombine outperforms all existing methods.** Several tools for batch correction of both flow and mass cytometry data have been published. We tested the performance of all maintained, peer-reviewed tools: CytoNorm, CytofRUV, CytofBatchAdjust, and

iMUBAC and compared their performance to cyCombine. To ensure a fair and broad comparison, we applied all tools to all the datasets used in the respective publications. As these tools have various limitations (e.g., designed to handle only one specific data type or condition, or designed to be dependent on technical replicates), each tool was tested only on datasets for which it was explicitly designed and tested by the authors. cyCombine was the only tool in the test that could handle every single dataset in full and showed superior performance for all of them when comparing the EMD reduction and MAD score (Fig. 5a, b). Selected density plots for the different tools and datasets are shown in

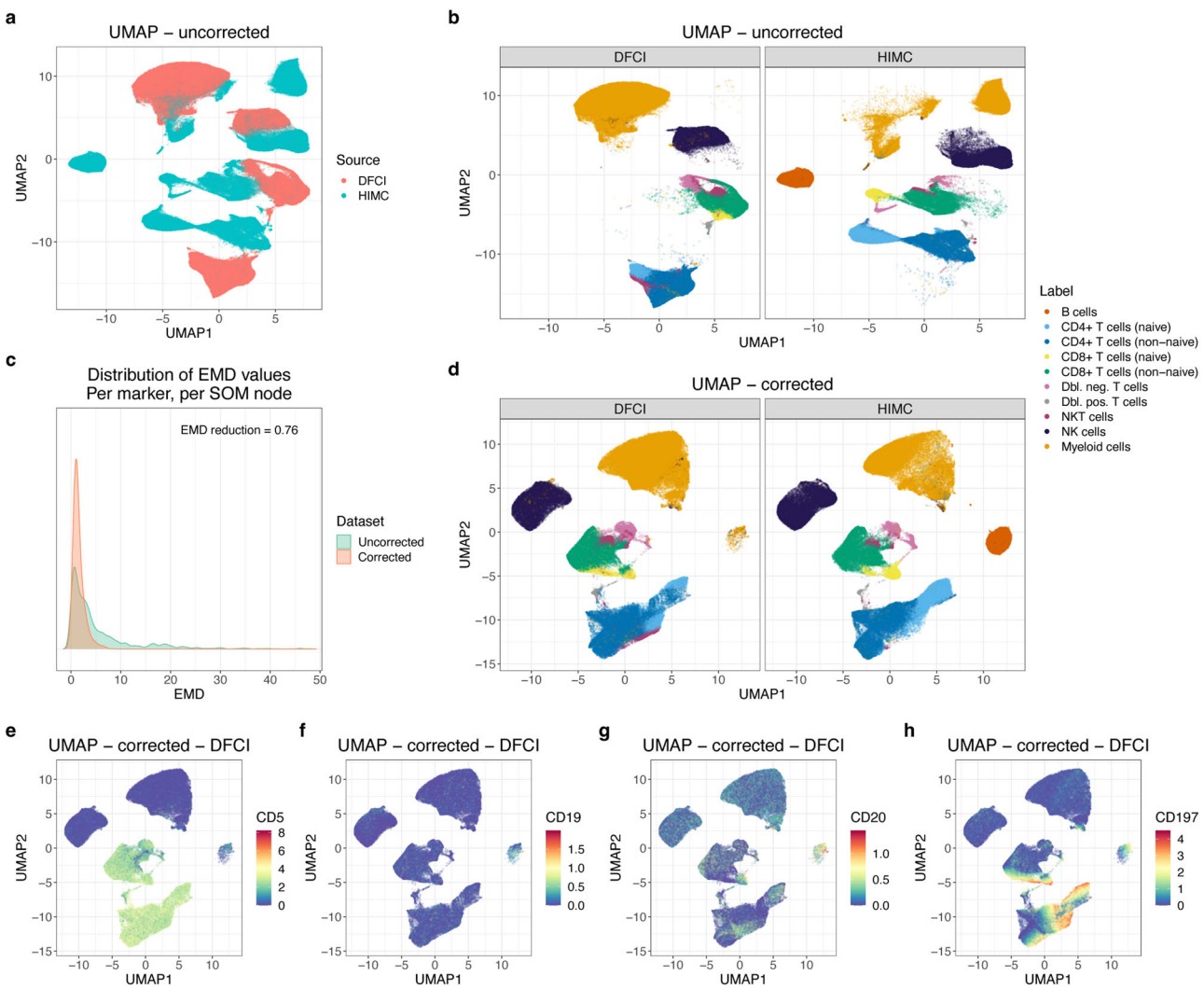

**Fig. 3 cyCombine rank-based batch correction for an HD sample from the Human Immune Monitoring Center (HIMC) dataset and an HD and a CLL sample from panel 1 of the Dana-Farber Cancer Institute (DFCI) data. a** UMAP for all cells from the two datasets based on expression of the 12 overlapping markers used for manual gating before batch correction. Colored by dataset. **b** Same as in **a**, but faceted by dataset and colored by manually assigned labels. **c** Earth mover's distance (EMD) density plots for uncorrected and corrected data, per marker, per self-organizing map (SOM) node. The EMD reduction was 0.76 and the MAD score was 0.04. **d** UMAP for all cells from the two datasets based on expression of the 12 overlapping markers used for manual gating after batch correction. Colored by manually assigned labels (assigned before correction). **e–h** Same as in **d**, DFCI, but colored by expression of CD5, CD19, CD20, and CD197 before batch correction.

Supplementary Figs. 8 and 9. Markers for the different datasets were selected such that they illustrate the performance differences between the benchmarked tools. One characteristic of the corrections by iMUBAC is a tendency to over-correct some batches, such that a peak is moved to become misaligned with the corresponding measurements in other batches. This is shown in Supplementary Fig. 8g, where the high-expression peak of CD4 in batch 2 is moved too far to the left, and in Supplementary Fig. 8k, n, where negative-value peaks are introduced by iMUBAC, but not by cyCombine. For CytoNorm, the changes between uncorrected and corrected are relatively small, but in some cases lower peaks in some batches seem to be moved slightly away from the zero-inflated distribution seen in uncorrected data, without a clear reason (Supplementary Fig. 9a, b). For CytofRUV, the MAD scores tend to be higher, reflecting a removal of biological variance. This is also shown in the density plots, e.g., in Supplementary Fig. 9a, f. Finally, CytofBatchAdjust appears to have a tendency to introduce extra peaks, which are not found in the uncorrected datasets (Supplementary Fig. 9d).

## Discussion

Deeper cytometric characterization of cell populations can have great implications, such as better diagnostics, development of novel therapeutics, and identification of important markers of immunity. However, a robust batch correction method is needed in order to fully realize the potential of single-cell cytometry. Correction of batch effects is often necessary to detect subtle biological variance in multi-batch experiments, and it is almost certainly a necessity for large-scale integration of data from different experiments.

In cyCombine, we handle cellular heterogeneity by applying careful overclustering of the data using a SOM. Co-clustering of data from all batches is enabled by an intermediary transformation of the expression values. The subsequent batch correction is performed using an empirical Bayes model, designed to reduce technical noise, while maintaining the biological signal. While others have previously used the EMD as a metric to measure the reduction in technical variance, we additionally describe the use of the MAD for quantifying the conservation of biological variance,

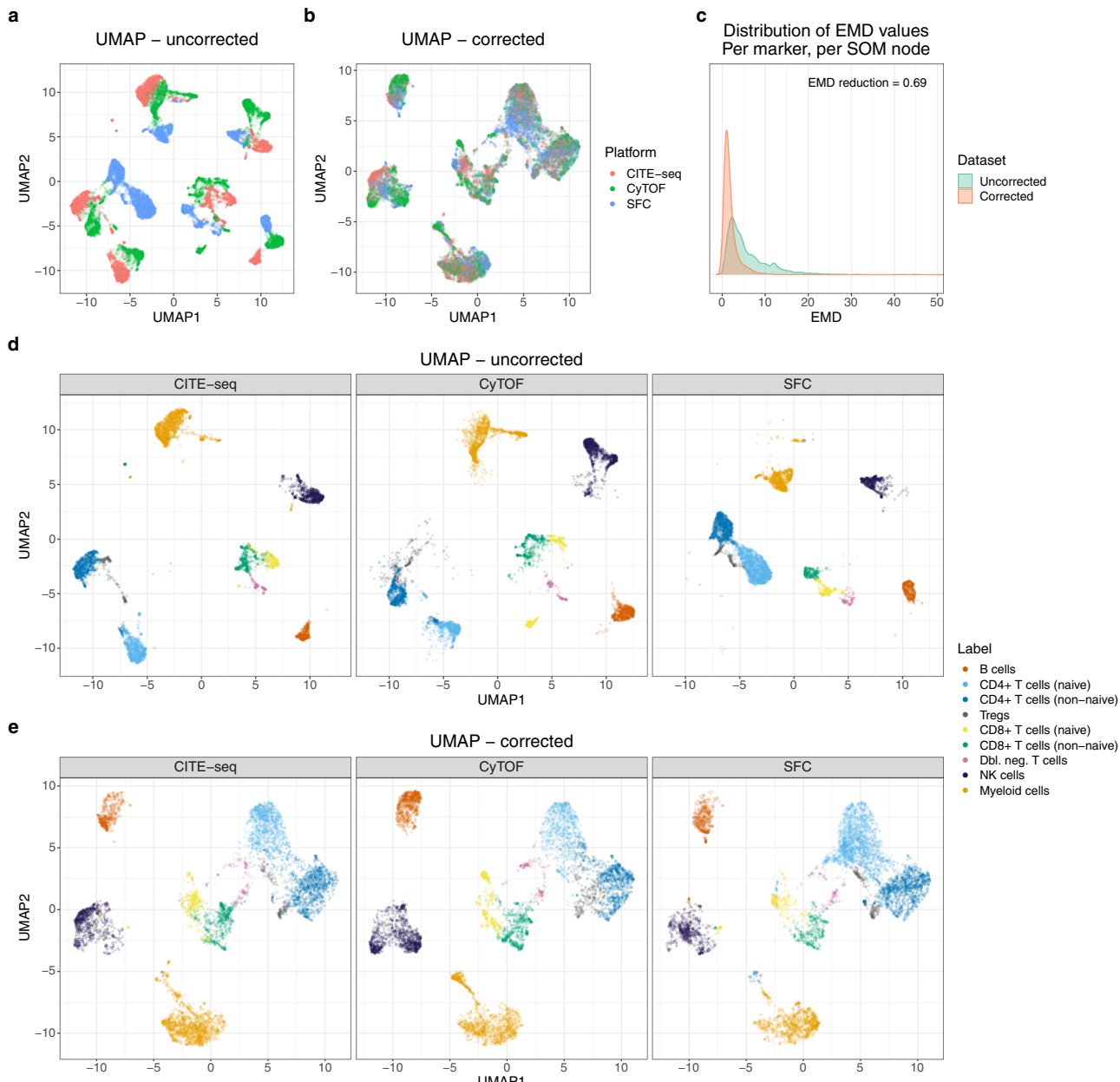

**Fig. 4 Cross-platform data integration. a** UMAP plot for uncorrected dataset consisting of 6776 cells from each of the CITE-seq, mass cytometry (CyTOF), and spectral flow cytometry (SFC) datasets. **b** UMAP for the cells of a after batch correction with cyCombine. **c** Density plots for earth mover's distances (EMD) calculated per marker, per SOM node for each of the pairwise comparisons between platforms. The self-organizing map (SOM) nodes used were those derived from corrected data. **d** The uncorrected UMAP faceted by technology and colored by manually assigned labels determined on each dataset separately before correction. **e** Same as in **d**, but for the corrected UMAP.

which is a feature that has been overlooked in the majority of previously published methods.

Using these metrics, we demonstrate that cyCombine batch correction is quantifiably more accurate than existing tools, and through analysis of three different biologically relevant datasets, we highlight the high degree of flexibility and robustness of our method: cyCombine is independent of technical replicates across batches and makes no assumptions about homology of marker expression distributions. It is largely insensitive to sample and batch sizes, as it handles batch correction for as few as eight cells in each SOM partition[11]. The SOM overclustering step ensures that both population abundances and cell phenotypes are retained, such that if batch effects are not present in a dataset, running the algorithm will not affect the expression values.

The primary limitations of cyCombine are inherited from ComBat, namely that batches and experimental conditions cannot be confounded. This means that at least one condition from each batch must be present in at least one other batch. Additionally, it is important to note that, while the cyCombine panel merging module enables imputation of non-overlapping features, batch correction is only possible for features present in all batches.

The accuracy of the imputations depends on the information content of the overlapping markers. Imputation is based on draws from multidimensional (kernel density estimated) distributions of the marker(s) to be imputed in the panel where their expressions were measured. In other words, the imputation is essentially a copy of the expression of the given marker(s) from highly similar

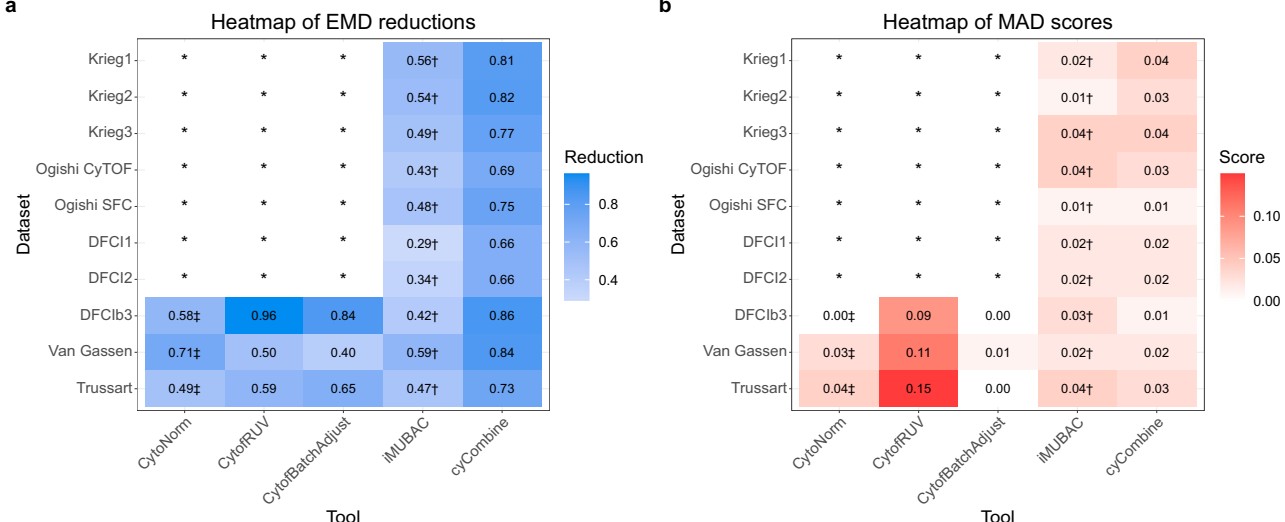

**Fig. 5 Performance evaluation of cyCombine and other previously published tools. a** Heatmap showing the earth mover's distance (EMD) reductions of the batch correction tools run on various datasets. A reduction of 1 means a complete elimination of EMD, 0 means no change in EMD. The best-performing setting was selected for each tool. **b** Heatmap showing the median absolute deviation (MAD) scores of the batch correction tools run on various datasets. A score of 0 means a complete preservation of the biological variance of all markers in all batches. The best-performing setting was selected for each tool. In both **a** and **b**, * denotes that the tool is dependent on technical replicates, which is not available in the dataset. † denotes that the tool is only applied for healthy donor samples and utilizes subsampling. ‡ denotes that the tool only corrects non-replicates and evaluations are performed on a subset of the full data.

cells from the marker-containing panel. This means that cell frequencies, marker distributions, and co-expressions are completely preserved. However, if the overlapping panel of markers is not able to accurately co-cluster cells expressing the markers to be imputed, the imputations will not be meaningful. As such, imputed marker expressions should generally only be used for visualization purposes, and we do not recommend basing differential expression analyses directly on imputed values as this can lead to inflated *p* values. Please refer to the panel merging vignette at https://biosurf.org/cyCombine for deeper discussion and thorough performance evaluation of cyCombine and other panel merging tools.

Both the challenge and the possibilities presented here become no less relevant when both the rate of growth and heterogeneity of cytometry data increases as new technologies become more prevalent. cyCombine scales linearly with the number of cells, and we envision that cyCombine will catalyze an increase of large-scale analyses of cytometry data. Of particular interest are applications such as harmonization of clinical cytometry data, which may enable better application of machine learning algorithms for diagnostics, for example by enabling faster detection of minimal residual disease in hematological cancers. A range of use cases, including code and in-depth discussions, are available in the cyCombine vignettes: https://biosurf.org/cyCombine.

## Methods

**The cyCombine package**. cyCombine was designed with protein expression-based cytometry data in mind, and the functions for data preparation are made to handle FCS files. cyCombine assumes that the data has already been pre-gated (i.e., beads, dead cells, doublets, debris, etc. have been removed). When using the built-in functions, the data will be ArcSinh-transformed with a cofactor of choice (recommended cofactors are 5 for CyTOF, 150 for flow cytometry, and 6000 for spectral flow cytometry). For CyTOF data, if counts are randomized, de-randomization is recommended[17]. However, the modules of cyCombine are not limited to data in FCS format, but are designed to work on any expression matrix that can be represented in an R data.frame—including CITE-seq protein expression data etc. cyCombine contains functions for importing FCS files, detection and correction of batch effects, plotting, evaluating batch correction, as well as performing panel merging. All functions are described in detail in the reference manual and the use case vignettes (https://biosurf.org/cyCombine).

**The cyCombine batch correction module**. cyCombine's batch correction module involves three separate steps: First, the expression of every marker is either Z-score normalized or converted to ranks, individually for each batch. Z-scoring is appropriate for similar datasets (e.g., multiple batches run on the same instrument with the same antibody clones and reporter molecules), whereas ranking tends to perform better for less similar datasets (e.g., data generated on different instruments, with different antibody-clones, different reporter molecules, or with different technologies). A SOM[12] is applied to the full normalized dataset. The grid size of the SOM should reflect the expected heterogeneity and result in a slight overclustering of the data. In cyCombine, the grid size defaults to 8 × 8, partitioning cells into 64 clusters. Then, the SOM node labels are assigned to the original expression value cells, a per cluster batch correction is applied using ComBat[11], and values are capped per-marker to the range of the input. The batch correction step can be performed with or without the use of a non-batch cofactor, e.g., phenotype or sample treatment. The cyCombine approach consequently allows for complex study designs, where not all conditions may be present in each batch, and where technical replicates were not included. It is possible to perform batch correction in studies with more than two conditions, and one may integrate different datasets with only one overlapping condition while accounting for this imbalance. The only requirement is that at least one condition from each batch is present in at least one other batch.

**Batch correction performance metrics**. In order to evaluate the performance of the methods, we primarily applied an approach based on the EMD strongly inspired by Van Gassen et al.[18]. The EMD has previously been suggested to be a good metric for comparing protein expression distributions[18,19]. Briefly, the EMD was used to compare the distribution of each marker within SOM nodes across batches. Generally, the SOM nodes were determined post-batch correction using 8 × 8 grids, and the labels were transferred to the uncorrected data so each cell had the same label in both the uncorrected and corrected data. For an in-depth discussion, see the performance benchmarking vignette at https://biosurf.org/cyCombine. The distributions were binned with bin size = 0.1, and the EMDs for every marker for each pairwise batch comparison were computed. These scores were determined for both the uncorrected and corrected data, removing those values where both had an EMD < 2. The EMD reduction is given as:

$$\mathrm{EMD}_{\mathrm{reduction}} = \frac{\sum_{i=1}^{n}\left(\mathrm{EMD}_{\mathrm{before}_i} - \mathrm{EMD}_{\mathrm{after}_i}\right)}{\sum_{i=1}^{n}\mathrm{EMD}_{\mathrm{before}_i}}, \qquad (1)$$

where $n$ is the total number of comparisons (number of SOM nodes times the number of markers times the number of pairwise batch comparisons). Furthermore, we have developed a score that reflects the amount of variance removed during a batch correction process. The score is based on the MAD and quantifies the variability of each marker in the dataset before and after correction. In practice, it is calculated very similarly to the EMD reduction: The MAD is calculated for the dataset after performing a SOM-based clustering, and is calculated per cluster, per marker, and per

batch. So, the MAD is calculated per batch, whereas the EMD calculations are performed for each pairwise batch–batch comparison. This means that the MAD score quantifies intra-batch effects of the correction, and the EMD reduction quantifies inter-batch effects. After calculating the MADs for both the corrected and uncorrected datasets, the MAD score is calculated as the median of the absolute difference in MAD per value:

$$\text{MAD}_{\text{score}} = \text{median}_{i=1}^{n}\left(\left|\text{MAD}_{\text{before}_i} - \text{MAD}_{\text{after}_i}\right|\right), \quad (2)$$

where $n$ is the total number of comparisons (number of SOM nodes times the number of markers times the number of batches). For an introduction to EMD reduction and MAD score, please see the performance benchmarking vignette at https://biosurf.org/cyCombine.

**The cyCombine panel merging module**. cyCombine also contains two functions for marker imputation. One function is designed with panel merging in mind and imputes the expression values of non-overlapping markers across two datasets. It works by first doing a SOM-based (defaults to an $8 \times 8$ grid) clustering of the datasets based on all of the overlapping markers. Then, for each cell in one of the datasets, the values for the missing markers are imputed by using the values from cells in the other dataset that fall within the same SOM node. The imputations are made by simulating a multi-dimensional kernel density estimate: Each cell's missing values are imputed by randomly drawing a cell from the other dataset and adding a Gaussian error, which is based on a draw from a Normal distribution with mean 0 and standard deviation corresponding to the bandwidth of each marker in the training population. However, if there are less than 50 cells from the other dataset within the SOM node, the values for the missing channels are set to NA as imputation would be unreliable.

The other function was made for salvaging a single channel within a dataset in selected batches. This can be useful in cases where one has a completely mis-stained marker in a single batch. It relies on the same principles, but instead of transferring information in one dataset to another, it utilizes intra-dataset batches.

**Chronic lymphocytic leukemia cohort**. CLL samples were obtained from the CLL Research Consortium (CRC) based at the University of California, San Diego, from patients who provided informed consent and as part of an institutional review board approved protocol. All samples were anonymized by the CRC. The dataset was generated at the DFCI and contained PBMC samples from 20 healthy donors (5 from DFCI and 15 from HemaCare) and samples from 56 patients with CLL. The latter were sampled at two distinct time points (T1 and T2), the mean time between T1 and T2 was 58.7 months (sd = 47.4 months), and T2 was obtained close to first treatment (mean = 4.5 months, sd = 10.4 months) (Fig. 6). For the 56 CLL patients, the mean age at diagnosis was 56.1 years (sd = 9.6 years), with healthy donors being age-matched (mean = 56.7 years, sd = 4.5 years). Serial samples from CLL patients along with PBMCs from healthy individuals were collected in accordance with the Declaration of Helsinki and written informed consent was obtained from all participants. No patients were compensated for their donation. A proportion of healthy donors samples were obtained for Hemacare and these donors were compensated for their time commitment during donation.

**Immunophenotyping CLL cohort using mass cytometry**. All patient and control PBMC samples were thawed in RPMI-1640 media (ThermoFisher) supplemented with 10% heat-inactivated FBS, sodium heparin (20 UI/mL) and 25 units/mL benzonase nuclease (Life Technologies and Sigma-Aldrich). Samples were subjected to B cell depletion using EasySep Human CD19 positive selection kit II (Stem Cell Technologies) before resuspension in RMPI and 10% FBS.

The samples were spun down and aspirated. Five micromolar of cisplatin viability staining reagent (Fluidigm) was added for two minutes and then diluted with culture media. After centrifugation, Human TruStain FcX Fc receptor blocking reagent (BioLegend) was used at a 1:100 dilution final in cell staining buffer (CSB) (PBS with 2.5 g/L bovine serum albumin and 100 mg/L of sodium azide, Sigma Aldrich) for 10 min followed by incubation with cell surface CyTOF antibody panels for 30 min (Supplementary Data 1). All CyTOF antibodies were obtained from the Harvard Medical Area CyTOF Antibody Resource and Core (Lederer Lab, Brigham and Women's Hospital, Boston, MA).

Sixteen percentage of stock paraformaldehyde (ThermoFisher Scientific) dissolved in PBS was used at a final concentration of 4% formaldehyde for 10 min in order to fix the samples before permeabilization with the FoxP3/Transcription Factor Staining Buffer Set (ThermoFisher Scientific). The samples were incubated with SCN-EDTA coupled palladium 20-sample barcoding reagents (Fluidigm) for 15 min, washed 3× in CSB, and then combined into a single 20 PBMC sample for subsequent staining. Conjugated intracellular CyTOF antibodies (Supplementary Data 1) diluted in the permeabilization buffer from the FoxP3/Transcription Factor Staining Buffer Set were added into each tube and incubated for 30 min. Cells were then fixed with 1.6% formaldehyde for 10 min.

The samples were processed in seven batches per antibody panel, each batch containing both control and patient samples. During sample processing, some samples were excluded due to dead cells or having too few cells to apply both panels. The final dataset has measurements for a total of 128 samples, all of which were included in the staining with panel 1, and 112 that were also stained with

panel 2. The 20 healthy donors were all stained with both panels. The CLL samples stained with panel 1 consisted of 52 samples at T1 and 56 (all patients) at T2. For panel 2, the numbers were 45 and 47, respectively. To identify single cell events, DNA was labeled for 20 min with an 18.75 μM iridium intercalator solution prior to acquisition. Samples were subsequently washed and reconstituted in cell acquisition solution in the presence of EQ Four Element Calibration beads (Fluidigm) at a final concentration of $1 \times 10^6$ cells/mL. Samples were acquired on a Helios CyTOF Mass Cytometer (Fluidigm).

**Analysis of CLL cohort mass cytometry data**. The raw FCS files were normalized to reduce signal deviation between samples over the course of multi-day batch acquisitions, utilizing the bead standard normalization method established by Finck et al.[20] as implemented in the premessa R package[21]. The normalized files were then compensated with a panel-specific spillover matrix to subtract cross-contaminating signals, utilizing the CyTOF-based compensation method established by Chevrier et al.[22] as implemented in CATALYST v. 1.12.2. These compensated files were then deconvoluted into individual sample files using a single-cell based debarcoding algorithm established by Zunder et al.[23] available in premessa v. 0.2.6. This was followed by pre-gating to live intact singlet cells using FlowJo version 10 (Tree Star Inc) as shown in Supplementary Fig. 10.

The pre-gated FCS files for each panel were read into R v. 4.0.0[24] using the cyCombine prepare_data function, using de-randomization and ArcSinh-transformation with cofactor = 5. The two panels consisted of a total of 6,027,290 and 6,831,388 cells. Subsequently, each panel was batch corrected using cyCombine with scaling and an $8 \times 8$ SOM grid using CLL/HD status as cofactor. After correction, all cells were clustered using an $8 \times 8$ SOM grid and the labels were transferred to the uncorrected data. The EMD was calculated for each marker comparing the batches and the EMD reductions and MAD scores between corrected and uncorrected data were determined for each panel. The data from the two panels was then co-batch corrected using the 15 overlapping markers with scaling and an $8 \times 8$ SOM grid maintaining CLL/HD status as cofactor but using panel as batch. After co-correction, the 40 (19 + 21) non-overlapping markers were imputed using an $8 \times 8$ SOM grid and the resulting datasets were combined to a single 55-marker dataset.

The 55-marker data was then clustered using a $10 \times 10$ SOM grid[12] and ConsensusClusterPlus v. 1.54.0[13] using 23 markers: CD3, CD4, CD8, CD45RA, CD45RO, CD197, CD127, CD25, CD5, CD19, CD20, CD56, CD16, CD33, CD14, HLA-DR, CD123, CD1c, CD1d, CD11c, CD11b, FCER1A, and CD34. The result was extracted for 45 meta-clusters, and each of these was manually annotated based on its marker expression. Among these clusters, there were eight pairs of clusters, which displayed highly similar expression patterns. Consequently, each of these sets were merged to a single final cluster, as previously described[25], leaving 37 clusters. Four of those clusters were labeled as either B cells (CD19+ CD20+) or CLL cells (CD19-lo CD20-lo CD5+), but because these populations can be considered cells that escaped the applied depletion, we removed those clusters from downstream analysis. Furthermore, four clusters displayed abnormal expression patterns, e.g., lack of lineage markers. When considering the mean viability stain for the clusters, it was observed that these four clusters all fell within the top-six highest values. This, together with the abnormal expression patterns, indicated that these clusters were composed of poor-quality cells, which we also excluded from further analysis. This left a final set of 29 populations and 10,719,711 cells to study.

Differential abundance testing was carried out using an approach presented by Weber et al.[26] (testDA_voom). Each test included individual false discovery rate (FDR)-correction for the populations included, but no correction was performed between tests. Instead, a FDR-threshold of 0.01 was used for significance. When relevant, the paired nature of the data was considered by using random effects. For differential expression testing within clusters, we analyzed the cell originating from each panel separately, meaning that no imputed values were included. The methodology for differential expression testing was also derived from the work by Weber et al.[26] (testDS_limma), in which medians serve as the foundation of the tests. Only markers not used for clustering were included in testing. Again, pairedness was considered when appropriate, and an FDR-threshold of 0.01 was used.

**HIMC healthy control sample**. A single healthy donor PBMC sample (Human Immune Monitoring Center (HIMC) healthy donor, ctrls-001, MATLAB-normalized) was downloaded from FlowRepository (ID: FR-FCM-ZYAJ) and pre-gated to live intact singlets in FlowJo version 10 (Tree Star Inc). The 174,601 cells were processed in R using cyCombine with de-randomization and ArcSinh-transformation with a cofactor = 5. For the integration with the CLL dataset, this was followed by manual gating to 10 cell types based on the lineage markers, CD3, CD4, CD8, CD14, CD19, CD20, CD33, CD45RA, CD56, CD161, CD197, and HLA-DR. Unlabeled cells (n = 615) were discarded. For the three-datatype integration, the pre-gating was followed by clustering to 20 meta-clusters using a $6 \times 6$ SOM[12] grid and ConsensusClusterPlus[13] based on expression of 11 markers overlapping with the healthy donor spectral flow cytometry (SFC) and CITE-seq sets (CD3, CD4, CD8a, CD14, CD16, CD19, CD25, CD45RA, CD56, CD127, and PD-1). These clusters were annotated manually based on protein expression levels, and 8932 cells were removed due to ambiguous expression patterns.

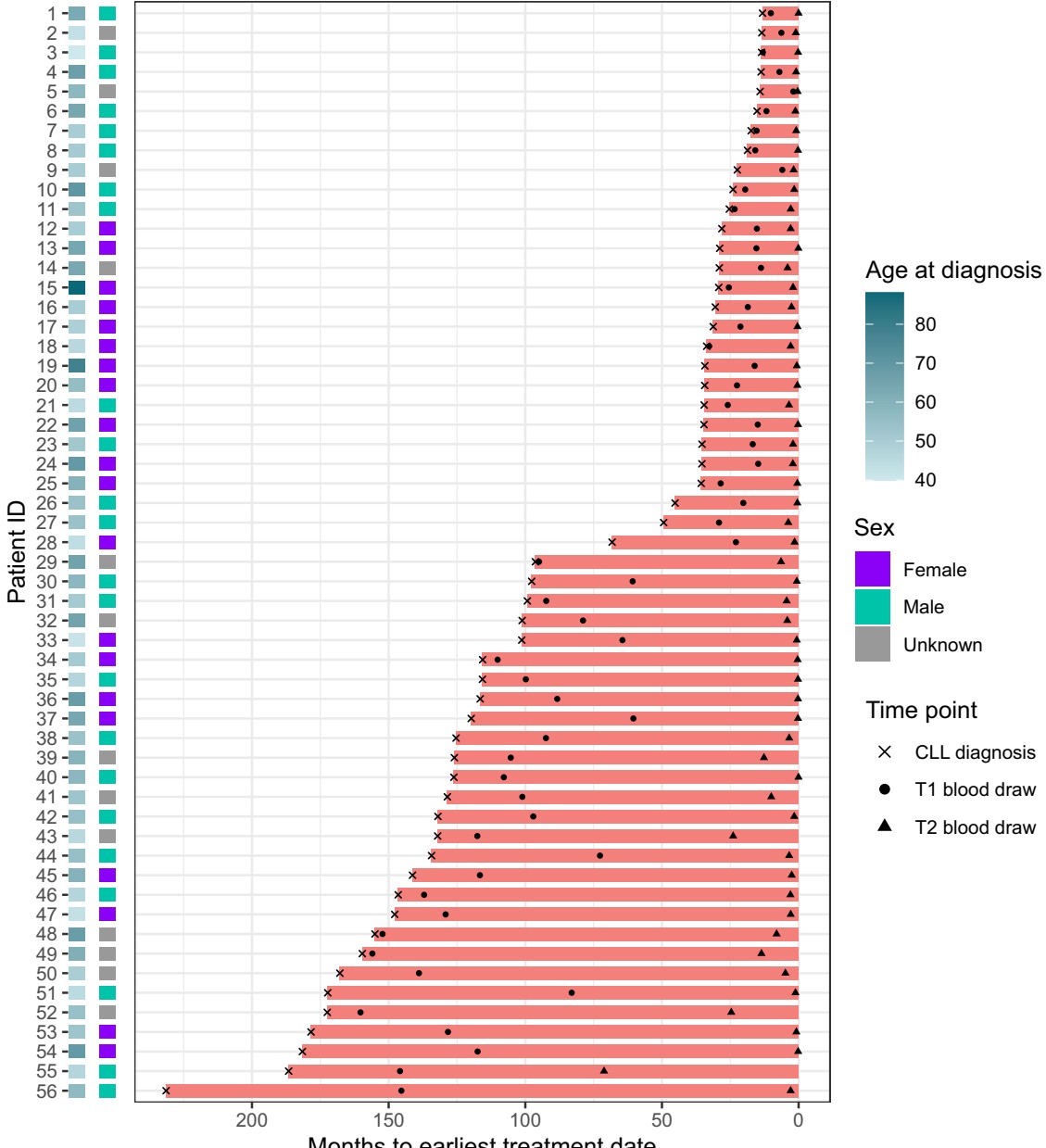

**Fig. 6 Chronic lymphocytic leukemia (CLL) cohort overview.** Time from CLL diagnosis to treatment initiation (months) for 56 CLL patients. Timing of blood draws for the time point 1 (T1) and time point 2 (T2) timepoints included in this study are indicated.

**Flow cytometry dataset**. The SFC dataset from Park et al.[5] was downloaded from FlowRepository (ID: FR-FCM-Z2QV). The dataset consists of samples from four healthy donor PBMCs, which were frozen and thawed, stained with 40 different antibodies in one panel, and analyzed using a 5-laser full spectrum flow cytometer (Cytek Biosciences Aurora).

Pre-processing was carried out in FlowJo version 10 (Tree Star Inc). The dataset was gated on lymphocytes, and singlets and non-debris were identified using forward and side-scatter. Dead cells were excluded using live/dead stains. Data from these gates were then exported in FCS format before further analysis in R: Using cyCombine, the data was loaded and transformed using ArcSinh with a cofactor = 6000. A single sample (donor 303444) with 582,005 cells was selected and clustered to 20 meta-clusters using a 6 × 6 SOM[12] grid and ConsensusClusterPlus[13] based on expression of 11 markers overlapping with the healthy donor CyTOF and CITE-seq sets. The clusters were annotated manually based on protein expression levels, and 21,307 cells were removed due to ambiguous expression patterns.

**Sequence barcoding-based dataset**. The filtered feature/cell matrix from the "10k PBMCs from a Healthy Donor—Gene Expression and Cell Surface Protein" dataset was obtained from the 10× website (https://support.10xgenomics.com/single-cell-gene-expression/datasets/3.0.0/pbmc_10k_protein_v3). This data was generated on the PBMCs of a single healthy donor stained with TotalSeq-B antibodies. It was sequenced on an Illumina NovaSeq and processed by Cell Ranger v. 3.0.0.

The TotalSeq expression matrix was processed in R using Seurat v. 4.0.0[27]. First, cells were filtered to maintain only those expressing between 200 and 2800 genes, having less than 10,000 detected RNA molecules and 20,000 detected protein molecules, and with a mitochondrial gene percentage below 10, leaving 6949 cells for analysis. The protein portion of the data was normalized, scaled, and dimensionality reduced to the 11 markers overlapping with the CyTOF and SFC datasets, before applying Louvain clustering at a resolution of 0.2. The 12 resulting clusters were manually annotated based on the expression levels of the 11 clustering proteins. Two clusters were considered to be doublets and excluded from the downstream integration, leaving 6776 cells.

**Integration of CLL and HIMC healthy donor sample**. For the integration with the HIMC healthy donor sample, two samples from the DFCI set (one CLL and one HD) from panel 1, batch 5 were selected (before any batch correction was applied) and manually gated to 10 cell types based on 12 lineage markers: CD3, CD4, CD8, CD14, CD19, CD20, CD33, CD45RA, CD56, CD161, CD197, and HLA-DR. Unlabeled cells (n = 4353) were considered to be representative of the low-quality

cells, and were discarded along with any cells labeled as B cells, since these were residual cells resulting from incomplete depletion. The HIMC sample was likewise gated to ten populations using the same 12 lineage markers. This resulted in a total of 352,210 cells, with 17 overlapping markers between the datasets (CD3, CD4, CD8, CD14, CD19, CD20, CD25, CD27, CD33, CD45RA, CD56, CD127, CD161, CD197, HLA-DR, ICOS, and PD-1). Datasets were batch corrected using cyCombine with an $8 \times 8$ SOM grid with the rank normalization method (and average ties method). Each set was considered a batch, and the HD/CLL status was used as a cofactor. The result of the batch correction was evaluated with the EMD reduction and MAD score as well as visual inspection of UMAP plots comparing the location of each cell type (which was assigned separately) across datasets.

**Integration of cross-platform datasets.** The HIMC CyTOF sample, the SFC sample, and the CITE-seq data were batch corrected together following the pre-processing described in the section for each set. Before batch correction, each set was downsampled to 6776 cells and to the 11 overlapping protein markers. This was followed by cyCombine batch correction with an $8 \times 8$ SOM grid with the rank normalization method (and average ties method). Each dataset was considered a batch and no cofactors were considered. The result of the batch correction was evaluated with the EMD reduction and MAD score as well as UMAP plots comparing the location of each cell type (which was assigned separately) across datasets.

**Benchmarking.** We compared the performance of the cyCombine batch correction module with four batch correction algorithms designed to work with mass cytometry data: CytoNorm[18], CytofRUV[28], iMUBAC[29], and CytofBatchAdjust[30]. Other tools exist, both developed for flow and mass cytometry, including gaussNorm and fdaNorm[31,32], which the authors state are no longer supported, and the tools cydar[33], BatchEffectRemoval[34], BatchEffectRemoval2018[35], SAUCIE[36], and swiftReg[37], which are not included due to either not being peer-reviewed, not being maintained, requiring a license, or being designed to work only on very specific cases, such as harmonizing two technical replicates. We tested each included tool on the datasets from the original publications and the set of datasets from other publications deemed to be suitable by the authors of each tool; i.e., some tools require technical replicates and not all datasets include these. Furthermore, we only tested each tool on datasets from platforms for which the use is demonstrated in the original publication. For tools with multiple tested settings, the setting with the best overall performance based on both the EMD reduction and MAD score was recorded.

All five included tools were run on the CyTOF datasets originally presented in the CytoNorm and CytofRUV papers, as well as the DFCI samples from batch 3 of both panels 1 and 2, where each panel was considered a batch. We will refer to these sets as the Van Gassen, Trussart, and DFCIb3 data, respectively. Additionally, we batch corrected six CyTOF datasets and one SFC set without technical replicates using cyCombine and iMUBAC. These datasets are the DFCI panel 1 and panel 2 sets, and five datasets presented in the iMUBAC article: Each of the three panels of the Krieg dataset, as well as a CyTOF and a SFC set originally generated for iMUBAC, which we refer to as Ogishi$_{CyTOF}$ and Ogishi$_{SFC}$. An overview is presented in Table 1. All CyTOF datasets were ArcSinh-transformed with a cofactor = 5 for processing with all tools.

The Van Gassen dataset[18] consists of 40 samples from two healthy controls. They comprise unstimulated and stimulated samples each run ten times (ten batches). Thirty-seven protein markers were measured. The Trussart dataset[28] consists of 24 samples from nine healthy controls (HCs) and three CLL patients, each run twice (two batches). Thirty-one protein markers were measured. The FCS files were pre-processed with bead normalization and debarcoding according to the script from the CytofRUV supplementary files (using CATALYST). The Krieg1, Krieg2, and Krieg3 datasets[38] comprise 30, 26, and 25 markers, and each contain 60 samples. They were, according to the original publication, processed as four experimental batches. Three conditions are considered: Healthy donors ($n = 20$), responders ($n = 22$), and non-responders ($n = 18$) to anti-PD-1 immunotherapy.

Each condition is included in each of the four batches. The dataset was pre-processed according to the instructions in the iMUBAC article: DNA and viability intercalators were used to exclude dead cells, doublets, and debris with the prepSCE function from iMUBAC. The Ogishi$_{CyTOF}$ dataset[29] contains measurements on 38 protein markers and consists of 57 samples in seven batches. A total of three conditions were included: Healthy ($n = 50$), MSMD ($n = 5$), and Salmonellosis ($n = 2$). Some of the healthy samples are biological replicates. The dataset was pre-processed according to the instructions in the iMUBAC article: DNA and viability intercalators were used to exclude dead cells, doublets, and debris. The Ogishi$_{SFC}$ dataset[29] measured 18 protein markers across 14 samples in two batches. A total of three conditions were included: Healthy donors ($n = 11$) and two types of autoimmune disease ($n = 1$ and $n = 2$). The dataset was pre-processed according to the instructions in the iMUBAC article: The viability stain was used to exclude dead cells and logicle transformation was used. The DFCI sets comprised two conditions: Healthy donors and CLL samples. As mentioned, the panel 1 data (DFCI1) had 36 measured markers, and the panel 2 data (DFCI2) had 34 markers. The DFCIb3 set consisted of the samples originating from batch 3 in each of the two panels, which had 15 overlapping markers. The DFCI samples were pre-processed as described above.

When running CytoNorm, we used FlowSOM clustering with a $10 \times 10$ grid and 25 final clusters (no downsampling). The batch effects were modeled using 101 quantiles. All protein markers were included. For the Van Gassen set, the 20 samples from healthy control 1 were used to model batch effects and the 20 samples from healthy control 2 were normalized. Evaluation of batch effect reduction was carried out using only the samples from healthy control 2. For the Trussart dataset, the CLL2 and HC1 samples were used as the technical replicates (training data). The remaining 20 samples were used as validation data and the evaluation of batch effect reduction was carried out using only the HC2-9, CLL1, and CLL3 samples. For the DFCIb3 set, the CLL_08_T1 and HD_05 samples were used as technical replicates, and the remaining 35 samples were used for evaluation. Corrected values were capped at 300 to avoid problems with very large values during evaluation.

For running CytofRUV, we used clustering with 20 clusters on lineage markers only (24 for Van Gassen, 19 for Trussart, and 12 for DFCIb3). All markers were corrected at varying values of k = {5, 10, 15, 20}. For the Van Gassen set, all healthy control 1 samples were used as technical replicates (two sets of ten samples each). For the Trussart set, the CLL2 and HC1 samples were used as the technical replicates, and for the DFCIb3 set, the CLL_08_T1 and HD_05 samples were used. All samples were included in the evaluation.

For running CytofBatchAdjust, all files were renamed according to the tool requirements. For Van Gassen, PTLG021 was used as the reference batch and the unstimulated healthy control 1 samples were used as anchors. We tested CytofBatchAdjust with method = {95p, SD, quantile} and transformation = {TRUE, FALSE}. For the Trussart set, HC1 was used as the anchor sample and RUV1b samples as reference batch, whereas DFCIb3 correction used HD_05 as the anchor and panel 1 as the reference batch. All markers were used for correction and all samples were used in evaluation. Corrected values were capped at 300 to avoid problems with very large values during evaluation.

iMUBAC was run largely according to the details in the original article. For all datasets, only healthy donors were included in correction, and downsampling to 200,000 cells for each batch was applied for all datasets, except for the Krieg3 dataset, for which we downsampled to 50,000 cells per batch, and the Ogishi$_{SFC}$ set, for which we downsampled to 500,000 cells per batch. For the Ogishi$_{CyTOF}$ set, only 47 local healthy donor samples were included as in the original publication (travel/family controls excluded). All evaluations were based solely on the downsampled datasets using all markers.

cyCombine was generally run on all available samples using the conditions stated in the presentation of each dataset. We ran cyCombine with norm_method = {scale, rank} on the full datasets with all markers.

**Table 1 Datasets used for benchmarking study.**

| Dataset | Instrument | Samples | Batches | Conditions | Cells (million) | Markers | FlowRepository ID | Originally used for tool |
|---|---|---|---|---|---|---|---|---|
| Van Gassen[43] | CyTOF2.0 | 40[a] | 10 | 2 | 6.2 | 37 | FR-FCM-Z247 | CytoNorm[18] |
| Trussart[28] | Helios | 24[a] | 2 | 2 | 8.6 | 31 | FR-FCM-Z2L2 | CytofRUV[28] |
| Krieg1[38] | Helios(2.1) | 60 | 4 | 3 | 1.1 | 30 | FR-FCM-ZY34 | iMUBAC[29] |
| Krieg2[38] | Helios(2.1) | 60 | 4 | 3 | 1.7 | 26 | FR-FCM-ZY34 | iMUBAC |
| Krieg3[38] | Helios(2.1) | 60 | 4 | 3 | 0.3 | 25 | FR-FCM-ZY34 | iMUBAC |
| Ogishi$_{CyTOF}$[29] | Helios | 57[a] | 7 | 3 | 12.4 | 38 | FR-FCM-Z3YK | iMUBAC |
| Ogishi$_{SFC}$[29] | Aurora | 14 | 2 | 3 | 9.7 | 18 | FR-FCM-Z3YL | iMUBAC |
| DFCI1 | Helios | 128 | 7 | 2 | 6.0 | 36 | FR-FCM-Z52G | cyCombine |
| DFCI2 | Helios | 112 | 7 | 2 | 6.8 | 34 | FR-FCM-Z52G | cyCombine |
| DFCIb3 | Helios | 39[a] | 2 | 2 | 1.9 | 15 | FR-FCM-Z52G | cyCombine |

[a]Counting replicates as distinct samples.

**Runtime and memory requirements**. We used the Ogishi$_{CyTOF}$ dataset comprising seven batches and 38 protein markers for testing the runtime and memory usage of the different tools. Several of the evaluated tools ran directly on FCS files; therefore, running these tools on a range of different sizes required storing downsampled versions of the original FCS files in new ones. This was done by loading the original FCS files, disregarding non-overlapping columns, sampling to the predefined sample sizes, and storing the resulting data in respective folders. By storing the data this way, it was ensured that all tools were run on the same data at each data size. The runtime and memory usage were measured for each tool for every sample size using the UNIX command time -v. The Maximum resident set size and the elapsed parameters in the output defined the memory usage and runtime, respectively. The test was performed on 40 cores (although none of the tools are fully parallelized, some sub functions are) on an HPE Apollo 2000 system with up to 192 GB PC4 2933 RAM. The standard laptop was a 2018 MacBook Pro with 16 GB 2400 MHz DDR4 memory and a 2.6 GHz 6-Core Intel Core i7 processor.

**Plots**. UMAPs were generated using uwot v. 0.1.9[39] on no more than approximately 500,000 cells (to avoid overcrowding the plots). Samples were downsampled if more cells were present, whereas all statistical analyses and clustering were done on the full datasets unless otherwise specified. Plots were generated using ggridges v. 0.5.2[40] and ggplot2 v. 3.3.3[41], and patchwork v. 1.1.1[42] was used for combining plots.

**Reporting summary**. Further information on research design is available in the Nature Research Reporting Summary linked to this article.

## Data availability

The DFCI CyTOF data generated in this study have been deposited in the FlowRepository database under accession code FR-FCM-Z52G. The HIMC CyTOF data used in this study are available in the FlowRepository database under accession code FR-FCM-ZYAJ. The Park flow data used in this study are available in the FlowRepository database under accession code FR-FCM-Z2QV. The van Gassen CyTOF data used in this study are available in the FlowRepository database under accession code FR-FCM-Z247. The Trussart CyTOF data used in this study are available in the FlowRepository database under accession code FR-FCM-Z2L2. The Krieg CyTOF data used in this study are available in the FlowRepository database under accession code FR-FCM-ZY34. The Ogishi CyTOF data used in this study are available in the FlowRepository database under accession code FR-FCM-Z3YK. The Ogishi flow cytometry data used in this study are available in the FlowRepository database under accession code FR-FCM-Z3YL. The CITE-seq data used in this study are available from the 10X genomics website [https://support.10xgenomics.com/single-cell-gene-expression/datasets/3.0.0/pbmc_10k_protein_v3].

## Code availability

The cyCombine R package is available on Github: https://github.com/biosurf/cyCombine/. Code to reproduce the analyses in this article is available at https://biosurf.org/cyCombine.

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

## Acknowledgements
This work was funded by the Independent Research Fund Denmark (grant 8048-00078B to LRO) and the National Institutes of Health (grants P30AR070253 and U01AI138318 to J.A.L.; P01CA206978 and UG1CA233338 to C.J.W.). S.H.G. was supported by the Kay Kendall Leukaemia Fund.

## Author contributions
L.R.O. and C.B.P. conceived the algorithm. C.B.P. and S.H.D. implemented the algorithm. C.B.P., L.R.O., M.B.B., M.D.L., and S.H.G. designed the use cases. C.B.P., S.H.D., S.H.G., and L.R.O. performed and interpreted the analyses. N.P., L.Z.R., T.J.K., J.N., J.A.L., S.H.G., and C.J.W. designed and generated the data for the chronic lymphocytic leukemia study. C.B.P., L.R.O., S.H.D., and S.H.G. wrote the manuscript. All authors edited the manuscript.

## Competing interests
C.J.W. holds equity in BioNTech, Inc; and receives research funding from Pharmacyclics. The remaining authors declare no competing interests.
