## [Peer Review File · Nature Communications]

cyCombine allows for robust integration of single-cell cytometry datasets within and across technologiesReviewers' Comments:

Reviewer #1:

Remarks to the Author:

This is a welcomed new computational tool to combine cytometry datasets. The data presented within the manuscript strongly illustrate the utility of the batch correction module, and the cross-platform integration module of CyCombine. The ability of the cytometry panel merging module using overlapping markers and imputation seems less robust, esp with the data shown in Fig 3, with B cells and myeloid cell 'confusion'. This is all fine to demonstrate the utility of the algorithm, but if one did not know that B cells were lacking in their dataset, they could have ended up with an incorrectly called myeloid cluster. It would be great if the authors could perform another analysis of, perhaps CyTOF data from overlapping multiple panels, using healthy human PBMC where there is known variability, but what the exact differences could be in marker expression are unknown. Finally, details on how many datasets can be combined at once using CyCombine are not clear. Details to users of the algorithm should be forewarned that the imputation could lead to false discovery as shown in Fig 3.

Reviewer #2:

Remarks to the Author:

Summary:

In this article, the authors describe a novel and innovative method, called, cyCombine, for normalization and integration of mass-cytometry data generated using different panels across different batches, labs, platforms, and technologies, an important innovation in the field. The integration across technologies (CyTOF, CITEseq, flow) is particularly exciting. The authors use well-accepted metrics such as EMD and MAD to demonstrate robustness of their method in integrating various protein-profiling datasets while retaining biological variance. The cyCombine convincingly outperforms other widely used software, addressing issues of previous methods such as not relying on technical replicates, and offers scalability for massive datasets. In addition to using publicly available datasets, the authors also generated a large dataset with experimental and technical biases from chronic lymphocytic leukemia (CLL) patients for establishing efficacy of cyCombine. This dataset additionally provides a resource for future benchmarking of such tools.

Major points:

1. The authors only show coarse cell type annotation for DFCI vs HIMC (~10 cell types), and CyTOF vs CITE vs SFC, as compared to the fine annotations defined on the dataset generated in-house (Suppl Fig 2, ~30 cell types). How robust is this approach in defining finer clusters?
2. In the performance evaluation in Figure 6, it would be nice to see the other tools assessed on more of the other datasets to increase overlap between different tools where possible. Perhaps the DFCI datasets could be evaluated using CytoNorm and CytofRUV since the panels have replicated samples?
3. The "panel merging" cyCombine module is the least convincing part of the paper. While imputation could work reasonably well for missing markers that may be reasonably redundant/overlapping in certain cell types (e.g. imputing CD19 if both panels have CD20), I would be concerned of its application for lineage markers that may be totally absent (i.e. imputing T cell markers in a panel that has zero T cell markers) or of activation/functional markers (e.g. Ki67, phospho-proteins, etc). A thorough discussion of this, and where panel merging is appropriate or not, is warranted.
4. It is difficult to see the extent of batch effect and subsequent correction using only the set of UMAPs provided as colored (and could be strongly affected by the ordering of how the plots are pointed due to the density of the points) in Figure 2 and Suppl Fig 1. To make it clearer to the reader, the following are recommended:
 - o a series of UMAPs that plot one batch in color at a time with the other batches plotted in gray (e.g. UMAP A shows batch 1 in color, remaining batches in gray; UMAP B shows batch 2 in color, remaining batches in gray; etc)

o quantifying batch composition per cluster (for example as stacked bars of batch composition per cluster)

Minor points:

1. The rationale behind the B cell and CLL cell depletion (and subsequent filtration of the residual) is not well explained. Additional details in the context of CLL would be useful in the methods or results section.
2. In Figure 6, are * and + correctly defined? I don't see any * shown (which are stated to denote the need for technical replicates), despite several tools in listed needing them
3. Scalability only goes up to 12 million cells – many datasets now have significantly more cells. Would the authors be able to test cyCombine for time/memory on a larger dataset?
4. There seems to be an erroneous "Figure 4" embedded in the methods ("time from CLL diagnosis"); there's already a figure 4 (and supp figure 4) that are different figures.
5. Figure legends: what are the dots colored by in 2f?

Reviewer #3:

Remarks to the Author:

It was with great pleasure I read the manuscript entitled: "Robust integration of single-cell cytometry datasets" by Pedersen and Dam et al. The paper addresses an important problem in the field of biology utilizing cytometry approaches (i.e., flow cytometry, mass cytometry, CITE-seq) of integrating datasets across different batches. The paper describes a new computational tool, inspired by DNA microarray data analysis, using an empirical Bayes method. This tool is potentially of broad interest for biologists working with big cytometry datasets, especially in the field of immunology. It offers advantages over other tools, by, for example, not relying on a reference sample. It shows promise in integrating datasets across different cytometry technologies as well, such as flow cytometry, mass cytometry, and CITE-seq. Congratulations to the authors for this great work, which is also very well written.

However, there are a few concerns that need to be addressed.

Major concerns:

1)

The extent of information loss after correction is unclear; this may particularly affect the preservation of rare subsets. It would be useful for the reader to understand to what degree rare subsets are preserved after the integration of the datasets.

- a. Initially, intra-sample heterogeneity is addressed by the grouping of similar cells using a self-organizing map (SOM) with an 8x8 grid. Since cytometry data contains many nonlinear patterns, important for accurate rare subset identification, how does this initial correction affect the preservation of rare subsets?
- b. In supplementary figure 1, multiple small clusters (in 1d) appear to disappear after correction (in 1e).
- c. Since it appears that in the manuscript, typically downsampled UMAPS are shown depicting multiple immune lineages, how does the uncorrected vs. corrected UMAP look like for zooming-in on an individual immune lineage, revealing more detail and thereby more rare cell clusters?

2) (related to point 1)

For data integration with partial overlapping markers, the extent of information loss is also currently not clear. Since this approach does not depend on reference samples, which is both an advantage and a disadvantage, a more detailed analysis would be useful. It is intuitive to assume that integrating the data with overlapping markers and imputing the non-overlapping markers can result in some degree

of information loss. CyTOFmerge by Abdelaal et al. (2019) initially calculates for a given dataset which set of markers best preserves the data heterogeneity so that in prospective studies, a partial overlapping panel can be designed with optimized markers as a core. The advantage of the current manuscript is the ability to merge pre-designed panels. However, it is currently not clear what the confidence level of imputation is for a given analysis and what potential information loss may have resulted from the correction.

a. An approach to elaborate on this could be to evaluate the quality of the imputed dataset compared to the original datasets. Compare the clusters of the original uncorrected datasets with the clusters of the imputed integrated dataset. Are there discrepancies between the two types of clustering?

b. It would be useful for the future users of this tool to have an idea of how successful the imputation was in terms of the preservation of information. If the overlap of n markers is only 5 vs. 10. vs. 15. vs. 20, how would this impact the analysis? Also, the nature of markers (canonical vs. noncanonical) would probably be a factor. The selection of shared markers is likely a strong dependency on success of imputation that should be addressed.

3) The section of cyCombine outperforms all existing methods shows interesting findings based on the EMD reductions and MAD scores. However, this section is an important part of the manuscript of putting this tool in comparison to existing tools and should be more elaborated. It would be useful if more visualizations are shown to compare this tool with existing tools. For example, UMAPS before and after correction comparing different datasets and different methods to help understand the additional benefit of the current tool better. The 2019 CyTOFmerge by Abdelaal et al. is also lacking, and it would be interesting to see this comparison.

Minor comments:

4) Since data heterogeneity a priori is often unknown, choosing the node grid size for the self-organizing map (SOM) may be difficult. How does the correction look like for different node grid sizes than 8×8 ?

5) How does the probability-based imputation compare with the knn-based imputation?

6) "The combined dataset was clustered based on a subset of 23 lineage markers using SOM12 and ConsensusClusterPlus13 to 45 meta-clusters, which were labeled manually, cleaned-up, and merged into a total of 29 clusters" Can the authors clarify what cleaned-up entails, and show the reasoning for the merging of the 45 meta-clusters to 29 clusters.

7) Consider putting supplementary figure 1 panel g and h in the main figure, as the correction looks very convincing there.

8) Figure 4: hard to see how the position of cells in uncorrected panel a change to new positions in corrected panel b. To understand the correction better, it would be interesting to see this change at the single cell level more clearly.

9) Figure 4: please show multiple major immune lineage markers for each of the three technologies before and after correction

10) Help the (naïve) reader to understand EMD reduction and MAD score better.

First and foremost, we would like to express our gratitude to the referees for investing time in reviewing our manuscript. We have received many great recommendations for additional analyses and discussions, and have done our best to add them to the manuscript. In some cases, we have chosen one of our online vignettes as the venue for the additional results and discussions. The primary reason is that the editorial restrictions on the publication format only allows for so many figures and so much text, but the format of the vignettes allows us to include many more examples than would be suitable for an article or its supplementaries, and it also allows us to organically include all the code necessary to reproduce the examples. If this raises any concerns, please do not hesitate to let us know and we will do our best to accommodate. We have provided point-by-point comments to all questions below, and redlined all changes to the manuscript and supplementary materials.

Reviewer #1 (Expertise: CyTOF, immune cells)

This is a welcomed new computational tool to combine cytometry datasets. The data presented within the manuscript strongly illustrate the utility of the batch correction module, and the cross-platform integration module of CyCombine. The ability of the cytometry panel merging module using overlapping markers and imputation seems less robust, esp with the data shown in Fig 3, with B cells and myeloid cell 'confusion'. This is all fine to demonstrate the utility of the algorithm, but if one did not know that B cells were lacking in their dataset, they could have ended up with an incorrectly called myeloid cluster.

Thank you for the positive feedback and the insightful suggestions. We agree that the imputation module was not as carefully discussed as the batch correction module in the original submission, and we have elaborated on this topic in the revised manuscript, the supplementary materials, and provided multiple detailed examples in the panel merging vignette (https://biosurf.org/cyCombine_panel_merging.html).

It would be great if the authors could perform another analysis of, perhaps CyTOF data from overlapping multiple panels, using healthy human PBMC where there is known variability, but what the exact differences could be in marker expression are unknown.

Using healthy human PBMCs to demonstrate the stability of imputations is a very good idea. To avoid crowding the manuscript, we have included such an example in the panel merging vignette (using two batches of healthy donors from the Ogishi dataset to compare batch correction and imputation).

Finally, details on how many datasets can be combined at once using CyCombine aren't clear. In theory, the upper limit is dictated only by the computational resources available to the user. We have elaborated on this in the revised manuscript.

Details to users of the algorithm should be forewarned that the imputation could lead to false discovery as shown in Fig 3.

We agree with the fact that imputation may lead to false inference in certain instances, and we have written this more directly in the revised manuscript in addition to the vignette. However, in Figure 3, imputation was not applied - only batch correction. The B cells that appear in the DFCl

sample in Figure 3D represent a mere 0.5% of the total population, and they appear to be CLL cells that escaped depletion rather than false positive per se, as discussed in the manuscript.

Reviewer #2 (Expertise: cytometry bioinformatics, systems immunology worked on CyTOF data)

Summary: In this article, the authors describe a novel and innovative method, called, cyCombine, for normalization and integration of mass-cytometry data generated using different panels across different batches, labs, platforms, and technologies, an important innovation in the field. The integration across technologies (CyTOF, CITEseq, flow) is particularly exciting. The authors use well-accepted metrics such as EMD and MAD to demonstrate robustness of their method in integrating various protein-profiling datasets while retaining biological variance. The cyCombine convincingly outperforms other widely used software, addressing issues of previous methods such as not relying on technical replicates, and offers scalability for massive datasets. In addition to using publicly available datasets, the authors also generated a large dataset with experimental and technical biases from chronic lymphocytic leukemia (CLL) patients for establishing efficacy of cyCombine. This dataset additionally provides a resource for future benchmarking of such tools.

Thanks a lot for your positive and very useful feedback. We have made the suggested changes and replied point-by-point to your comments below.

Major points:

1. The authors only show coarse cell type annotation for DFCI vs HIMC (~10 cell types), and CyTOF vs CITE vs SFC, as compared to the fine annotations defined on the dataset generated in-house (Suppl Fig 2, ~30 cell types). How robust is this approach in defining finer clusters? The clustering and cell type annotation for the cross-study examples (DFCI vs. HIMC and CyTOF vs. CITE vs. SFC) was limited by the overlapping markers, which in this case was only 11. For an integration of the CyTOF vs SFC datasets, we have in total 26 overlapping markers, which allows for a deeper characterization as requested. As shown below, this allowed us to identify a larger number of subpopulations, which are well-conserved when considering the data from each platform pre- and post-batch correction. To avoid crowding the manuscript with a lot of similar examples, we added the analysis and a thorough discussion in a dedicated usage vignette https://biosurf.org/cyCombine_Spectralflow_CyTOF.html

2. In the performance evaluation in Figure 6, it would be nice to see the other tools assessed on more of the other datasets to increase overlap between different tools where possible. Perhaps the DFCI datasets could be evaluated using CytoNorm and CytofRUV since the panels have replicated samples?

This is a good point and we have added analysis using batch 3 from both panels 1 and 2 (based on the 15 overlapping markers between the panels) to Figure 5. This means that there are essentially only replicates in the data (except for the single sample which was only included in panel 1). We selected batch 3 since it displayed the strongest batch effects out of the seven, although these were still relatively minor.

A couple of caveats of this analysis to consider: Firstly, the other tools are not geared for correction of samples, for which the panels are not identical. In the case of the DFCI dataset, we have both generally different panels, but also a marker included in both panels, that switches metal (HLA-DR). Because the tools read samples directly from the FCS format, we could only address these issues by either altering the actual FCS files, or editing the source code of the tools (which we are very hesitant to do) to only consider overlapping markers. Secondly, all the two-panel samples are technical replicates, which is a very artificial scenario and thus not ideal for testing batch correction performance (which is also true for the Van Gassen and Trussart sets). Biological variance between technical replicates is generally expected to be close to zero, making it a relatively simple problem to solve (variance is essentially only subject to removal, meaning that the challenges of maintaining biological variance are not applicable). We have added these points to the Supplementary Discussion.

3. The “panel merging” cyCombine module is the least convincing part of the paper. While imputation could work reasonably well for missing markers that may be reasonably redundant/overlapping in certain cell types (e.g. imputing CD19 if both panels have CD20), I would be concerned of its application for lineage markers that may be totally absent (i.e. imputing T cell markers in a panel that has zero T cell markers) or of activation/functional markers (e.g. Ki67, phospho-proteins, etc). A thorough discussion of this, and where panel merging is appropriate or not, is warranted.

This is a very good point, and we completely agree that (all) imputation should be used with caution. In this case, careful consideration of the set of overlapping markers is paramount to performance. We have elaborated on the advantages and shortcomings of panel merging in the manuscript discussion and updated the panel merging vignette with additional examples. We have also updated the R package to display a message cautioning users to analyze imputed values directly (we do not use imputed values in our statistical analyses - we only use them for visualization purposes).

4. It is difficult to see the extent of batch effect and subsequent correction using only the set of UMAPs provided as colored (and could be strongly affected by the ordering of how the plots are pointed due to the density of the points) in Figure 2 and Suppl Fig 1. To make it clearer to the reader, the following are recommended:

- o a series of UMAPs that plot one batch in color at a time with the other batches plotted in gray (e.g. UMAP A shows batch 1 in color, remaining batches in gray; UMAP B shows batch 2 in color, remaining batches in gray; etc)
- o quantifying batch composition per cluster (for example as stacked bars of batch composition per cluster)

We have added the suggested plot with grey “background” coloring to the supplementary figures (Supp. Figure 2).

The per cluster composition does not really change with batch correction (as indicated by supplementary figure 2), and we have been unable to convince ourselves that this would clarify the extent of batch effect/correction, even if it did. It is natural that composition varies from sample to sample (and thus there is no right or wrong batch composition). We apologize if this is not what you meant, and if you are not satisfied with our reply, please clarify and we will be happy to address this point.

Minor points:

1. The rationale behind the B cell and CLL cell depletion (and subsequent filtration of the residual) is not well explained. Additional details in the context of CLL would be useful in the methods or results section.

Fair point. We have elaborated on the rationale behind depletion being used for this particular study in the results section “cyCombine enables large-scale integration of multi-batch, multi-panel cytometry data”, and have further mentioned the arguments in relation to clean-up of the residual in the methods section.

2. In Figure 6, are * and + correctly defined? I don’t see any * shown (which are stated to denote the need for technical replicates), despite several tools in listed needing them

We have increased the size of the *’s, which were indeed very small in the original version. The figure is now ‘Figure 5’, since the numbering has been corrected.

3. Scalability only goes up to 12 million cells – many datasets now have significantly more cells. Would the authors be able to test cyCombine for time/memory on a larger dataset?

Because cyCombine scales linearly (both in time and memory) for all tested data sizes, the computational requirements for larger sets can be extrapolated directly from our existing analysis. We have elaborated on this in the revised manuscript.

4. There seems to be an erroneous “Figure 4” embedded in the methods (“time from CLL diagnosis”); there’s already a figure 4 (and supp figure 4) that are different figures.

Thank you for pointing out this mistake. we have corrected this in the revised manuscript.

5. Figure legends: what are the dots colored by in 2f?

The dots were colored by patient, but it actually is unnecessary. We have adjusted to use a single color.

I found this paper very impressive, thorough, and an important addition to the field. We have been working with several of the existing batch correction algorithms available for CyTOF data (CytofRUV, CytoNorm), and are pleased to see that cyCombine addresses several of their limitations. Would recommend this paper for publication (and am looking forward to giving it a try on some of our data!)

Thanks a lot - we really appreciate that you took the time to review the manuscript, and we are looking forward to hearing from users to keep updating and improving the cyCombine package.

Reviewer #3 (Expertise: high dimensional cytometry data analysis, junior)

It was with great pleasure I read the manuscript entitled: “Robust integration of single-cell cytometry datasets” by Pedersen and Dam et al. The paper addresses an important problem in the field of biology utilizing cytometry approaches (i.e., flow cytometry, mass cytometry, CITE-seq) of integrating datasets across different batches. The paper describes a new computational tool, inspired by DNA microarray data analysis, using an empirical Bayes method. This tool is potentially of broad interest for biologists working with big cytometry datasets, especially in the field of immunology. It offers advantages over other tools, by, for example, not relying on a reference sample. It shows promise in integrating datasets across different cytometry technologies as well, such as flow cytometry, mass cytometry, and CITE-seq. Congratulations to the authors for this great work, which is also very well written.

Thank you for taking the time to provide excellent suggestions for the improvement of our manuscript. We have addressed each of your points below.

However, there are a few concerns that need to be addressed.

Major concerns:

1)

The extent of information loss after correction is unclear; this may particularly affect the preservation of rare subsets. It would be useful for the reader to understand to what degree rare subsets are preserved after the integration of the datasets.

a. Initially, intra-sample heterogeneity is addressed by the grouping of similar cells using a self-organizing map (SOM) with an 8x8 grid. Since cytometry data contains many nonlinear patterns, important for accurate rare subset identification, how does this initial correction affect the preservation of rare subsets?

As a general feature of SOMs, the detection of rare clusters may indeed be affected by the grid size. In cyCombine, an 8x8 grid is applied by default as this captures rare populations in the vast majority of cases we have tested. The grid size is adjustable if more or less heterogeneity is anticipated. Generally speaking, increasing the grid size beyond an already good size won't change performance in any significant way (but it will increase runtime), whereas a grid size that is too small may lead to loss of rare subsets in the post-correction dataset. As in any analysis involving a clustering step, we would recommend initial explorations of all datasets to establish an idea of the heterogeneity in the data, and then set the grid size accordingly when running cyCombine. We added this recommendation and a detailed discussion of how to do this in practice to the vignette (https://biosurf.org/cyCombine_CyTOF_1panel.html), and also illustrate the effects of choosing different grid sizes with multiple examples.

b. In supplementary figure 1, multiple small clusters (in 1d) appear to disappear after correction (in 1e).

Good point. In order to trace the cells from the small clusters in Supplementary Figure 1d+e, we have made an additional Supplementary Figure 3 where we show that the smaller clusters move around in UMAP space (as expected when re-running the dimensionality reduction after correcting values), but that they are generally conserved.

c. Since it appears that in the manuscript, typically downsampled UMAPs are shown depicting multiple immune lineages, how does the uncorrected vs. corrected UMAP look like for zooming-in on an individual immune lineage, revealing more detail and thereby more rare cell clusters?

Downsampling was only used to avoid over-cluttering the UMAP plots. All clustering and statistical analyses were done on the full dataset. Of course, you are right that very rare cell types are at risk of being omitted from the visual representation when downsampling, but they were not omitted from any clustering, statistical analysis, nor interpretation. We have clarified this in the manuscript. Below we show UMAPs including *all* cells originating from panel 1, which were labeled as CD8+ T and NKT cells in the final clustering. Each row shows a batch and the three columns show the UMAP layouts for the uncorrected, corrected (panel 1 only), and corrected (together with panel 2) + imputed sets, respectively. Labels are those from the final clustering based on 23 lineage markers from the fully integrated set, and the UMAP layout is based on expression of the 36 panel 1 markers. When we visually inspect the plots below, we hope you agree that nothing indicates that downsampling distorts the visual representations to a worrying degree.

UMAPs - Panel 1 cells, 36 markers

2) (related to point 1)

For data integration with partial overlapping markers, the extent of information loss is also currently not clear. Since this approach does not depend on reference samples, which is both an advantage and a disadvantage, a more detailed analysis would be useful. It is intuitive to assume that integrating the data with overlapping markers and imputing the non-overlapping markers can result in some degree of information loss.

We have added two new examples to our panel merging vignette (https://biosurf.org/cyCombine_panel_merging.html), and we have also elaborated on these points in the discussion section of the manuscript. In the vignette, one example directly compares the results from batch correction on a large set of overlapping markers to batch correction on a smaller set and performing imputation for the remaining markers. In this example, it is directly seen how information is passed through the different modules of cyCombine.

CyTOFmerge by Abdelaal et al. (2019) initially calculates for a given dataset which set of markers best preserves the data heterogeneity so that in prospective studies, a partial overlapping panel can be designed with optimized markers as a core. The advantage of the current manuscript is the ability to merge pre-designed panels. However, it is currently not clear what the confidence level of imputation is for a given analysis and what potential information loss may have resulted from the correction.

You are correct that CyTOFmerge and the merging module of cyCombine differs on this key point, which is also why we didn't include a direct comparison with CyTOFmerge in the paper - it simply would not be fair to evaluate a tool's performance on a problem for which it wasn't designed to solve.

Regarding the confidence levels of imputation, you are correct that we did not explicitly calculate this, for which there are multiple reasons. First of all, imputations are only used for co-visualizing two datasets with non-overlapping markers. We do not perform any statistical analysis directly on the imputed values, and do not recommend users to do so (we also added a message to the script warning users not to directly analyze imputed values). We have added a more detailed discussion of this in the manuscript and the panel merging vignette. The second reason for not diving deeper into the accuracy of the imputations, is that they are, in a sense, "perfect". This is because we are imputing the expression of markers based on the expression patterns of highly similar cells in a technical replicate. The imputation is based on a multidimensional density draw, meaning that we essentially replicate the expression of the markers in the cells of the "training" set, which, quite naturally, will be extremely similar to the expression in the cells of the "test" set as these are technical replicates. So in this sense the imputed values are actually "perfect", but they are of course also perfectly uninformative in the sense that they do not supply any new information - they merely fill in blanks without adding noise (the very essence of imputations). This allows us to make one nice big plot using a method that does not handle missing values (UMAP, in this case). One should not impute markers from other samples (a worst case example would be imputing markers in a patient sample based on expression in a healthy donor sample), nor should one attempt to impute markers from a very small training set

where the heterogeneity of the missing values is not captured. We have elaborated extensively on this in the panel merging vignette (https://biosurf.org/cyCombine_panel_merging.html).

a. An approach to elaborate on this could be to evaluate the quality of the imputed dataset compared to the original datasets. Compare the clusters of the original uncorrected datasets with the clusters of the imputed integrated dataset. Are there discrepancies between the two types of clustering?

As discussed above, this would merely replicate the training set and the new clustering will look exactly the same (insofar as there are enough cells in the training set to capture the heterogeneity). However, this point is addressed in Supplementary Figure 3 (generated to answer a related question by Reviewer #1), which shows the clusters obtained from the fully corrected and imputed dataset in the UMAP layouts for each of the analysis stages in the CLL dataset. In these plots, each UMAP is generated using all markers available for the given set - i.e. the layout for panel 1 (a-b) is based on the 36 markers in that panel. If we consider panel 1 (a-b), we observe that the myeloid cell clusters are not well-separated. This is due to the lack of multiple myeloid markers in the panel, and as such, higher-resolution clustering of these cell populations is not possible for the panel 1-cells unless imputation is applied. A similar pattern is observed for the panel 2 data (c-d), when looking at the CD8+ T cell clusters - at least when considering the EM and TEMRA populations. As such, when we consider the clusters that could be obtained using the original uncorrected datasets vs. the “final” clusters based on the integrated and imputed data, there are natural discrepancies due to the different available markers. However, we also believe that the UMAPs in Supplementary Figure 3 depict that most of the clusters are indeed stable between the uncorrected and integrated datasets. We have added these points to the Supplementary Discussion.

b. It would be useful for the future users of this tool to have an idea of how successful the imputation was in terms of the preservation of information. If the overlap of n markers is only 5 vs. 10. vs. 15. vs. 20, how would this impact the analysis? Also, the nature of markers (canonical vs. noncanonical) would probably be a factor. The selection of shared markers is likely a strong dependency on success of imputation that should be addressed.

This is something which we have discussed extensively while developing cyCombine. We concluded that it is essentially not feasible to do a thorough test of all potential imputations - not even for subsets such as 5, 10, 15, 20 as suggested. The reason is, that the number of imputed markers alone is not enough to draw conclusion on accuracy - rather it comes down to whether the heterogeneity of the missing markers is captured in the “training” portion of the dataset, and whether the overlapping markers allow for a clustering that captures the similarities in the non-overlapping. E.g. imagine trying to impute all T cell markers, from a “training” set that contained no T cells (or no T cells discernible by clustering if the relevant markers were missing) - this couldn't be done. This means that a thorough test would be done on every combination of k markers drawn from the total set of n markers. This has a time complexity of $O(\binom{n}{k})$ for $n=25$ and $k=[5,10,15,20]$ (resulting in approximately 6.6 million comparisons), rendering the test *extremely* computationally demanding, and the results unmanageable to interpret. Instead, we have illustrated these points with ‘easy-to-impute’ and ‘hard-to-impute’ examples in the vignette,

and elaborated the discussion of the limitations of imputation in the manuscript. We hope this addresses the concern.

3) The section of cyCombine outperforms all existing methods shows interesting findings based on the EMD reductions and MAD scores. However, this section is an important part of the manuscript of putting this tool in comparison to existing tools and should be more elaborated. It would be useful if more visualizations are shown to compare this tool with existing tools. For example, UMAPs before and after correction comparing different datasets and different methods to help understand the additional benefit of the current tool better. The 2019 CyTOFmerge by Abdelaal et al. is also lacking, and it would be interesting to see this comparison.

This is a very good point - since one of our own key results are highlighted with dimensionality reduction and density plots, it is reasonable to add such visualizations for the other tools as well. We have added a discussion of these results to the revised manuscript and show the results in Supplementary Figures 8 and 9. Comprehensive analyses and discussions have also been added to the benchmarking vignette (https://biosurf.org/cyCombine_benchmarking.html), where we exemplify with UMAPs and density plots for all five tools on the Van Gassen dataset (if we were to show the density plots for each correction with all tools, we would have a total of 829 density plots, which would each contain both the uncorrected and corrected densities per batch). Furthermore, as there are 29 runs in total, for which we could show uncorrected and corrected UMAPs, it would likely be hard to grasp for the reader. We hope you agree that our selection of markers effectively illustrates the point.

Regarding CyTOFmerge: the merging module of cyCombine differs from this tool in that CyTOFmerge is designed to first detect the most informative markers, such that the panels can be designed to overlap with these channels. However, in many real world applications, the panels are designed and the samples run, prior to addressing merging. cyCombine is designed to work (almost) regardless of how the panels were designed, which means that while CyTOFmerge and cyCombine seemingly solve a very similar problem, they are based on different premises for data acquisition. This is why we didn't include a direct comparison with CyTOFmerge in the manuscript - it simply would not be fair to evaluate a CyTOFmerge's performance on a problem for which it wasn't designed to solve. We have discussed these key differences between the two tools in the updated panel merging vignette (https://biosurf.org/cyCombine_panel_merging.html), as we feel that the vignettes are better suited for these more theoretical discussions. The performance of these two imputation approaches is not directly compatible with the batch correction benchmark, which relies on the EMD reduction and MAD score.

Minor comments:

4) Since data heterogeneity a priori is often unknown, choosing the node grid size for the self-organizing map (SOM) may be difficult. How does the correction look like for different node grid sizes than 8x8?

As in any analysis involving a clustering step, we would recommend initial explorations of all datasets to establish an idea of the heterogeneity in the data, and then set a grid size that likely

captures all expected populations when running cyCombine. Decreasing the grid size means potentially missing rare populations in the initial clustering and also typically lowers the performance. Increasing the grid size to, for example, 12x12 or 16x16 doesn't affect performance in any meaningful way (we would always recommend an over-clustering if in doubt), but runtimes will increase. In all our tests of cyCombine on PBMC with standard phenotyping panels, 8x8 performs robustly. We added recommendations and a short discussion of how to do this in practice to the manuscript and a full analysis of different grid sizes to the batch correction vignette (https://biosurf.org/cyCombine_CyTOF_1panel.html).

5) How does the probability-based imputation compare with the knn-based imputation?

Our panel merging vignette (https://biosurf.org/cyCombine_panel_merging.html) has been updated to elaborate on this matter. Essentially, what was observed in our tests for cyCombine vs. CyTOFmerge was that the knn + median-based approach (CyTOFmerge) works reasonably well when considering unimodal distributions, but fails to grasp bimodality and often truncates distributions. Since informative/interesting markers are actually often those with larger variance, knn is suboptimal compared to probability-based imputation. In fairness, the CyTOFmerge tool was not designed to handle bimodal distributions, as the tool has an initial step that allows the user to design the panels selecting low variance markers for the non-overlapping portion.

6) "The combined dataset was clustered based on a subset of 23 lineage markers using SOM¹² and ConsensusClusterPlus¹³ to 45 meta-clusters, which were labeled manually, cleaned-up, and merged into a total of 29 clusters" Can the authors clarify what cleaned-up entails, and show the reasoning for the merging of the 45 meta-clusters to 29 clusters.

We have elaborated on the clean-up and merging in the methods section. Briefly, we labeled clusters based on their expression of the 23 lineage markers. In some cases, two clusters had very similar expression patterns for all of the major markers, and in those cases, we decided to use the same label for two clusters - effectively merging them. The clean-up was an extra step to get rid of any B/CLL cells that escaped the magnetic depletion.

7) Consider putting supplementary figure 1 panel g and h in the main figure, as the correction looks very convincing there.

We appreciate the suggestion, but we have decided to keep the figures as they were, since we believe that Figure 2a-b provides convincing results for the main figure. In addition, this means that all UMAPs in Figure 2 are based on the same part of the 23 lineage markers used in clustering, whereas Supp. Figure 1 contains UMAPs including all available markers for each set.

8) Figure 4: hard to see how the position of cells in uncorrected panel a change to new positions in corrected panel b. To understand the correction better, it would be interesting to see this change at the single cell level more clearly.

Great suggestion. We have added figures showing uncorrected data with labels (Figure 4d).

9) Figure 4: please show multiple major immune lineage markers for each of the three technologies before and after correction

Good suggestion. We have added this as Supplementary Figure 6 for six of the markers on un-faceted plots and for all eleven markers with faceted plots in the vignette: https://biosurf.org/cyCombine_CITEseq_Spectral_CyTOF.html.

10) Help the (naïve) reader to understand EMD reduction and MAD score better.

We have elaborated on this in our benchmarking vignette (https://biosurf.org/cyCombine_benchmarking.html), where we describe the EMD reduction and MAD score, and discuss their strengths and limitations. We have added a reference to this in the manuscript.

Reviewers' Comments:

Reviewer #1:

None

Reviewer #2:

Remarks to the Author:

The authors have provided satisfactory responses to my comments and I have no further concerns.

Reviewer #3:

Remarks to the Author:

Thank you for the elaborate reply.

All concerns have been addressed, and I fully support the publication of this work that I am sure will gain great attraction.

Reviewer #2 (Remarks to the Author):

The authors have provided satisfactory responses to my comments and I have no further concerns.

Reviewer #3 (Remarks to the Author):

Thank you for the elaborate reply.

All concerns have been addressed, and I fully support the publication of this work that I am sure will gain great attraction

We would like to thank all reviewers for their hard work and excellent comments on our work.